# DAdaQuant: Doubly-adaptive quantization for communication-efficient Federated Learning

## Abstract

Federated Learning (FL) is a powerful technique for training a model on a server with data from several clients in a privacy-preserving manner. In FL, a server sends the model to every client, who then train the model locally and send it back to the server. The server aggregates the updated models and repeats the process for several rounds. FL incurs significant communication costs, in particular when transmitting the updated local models from the clients back to the server. Recently proposed algorithms quantize the model parameters to efficiently compress FL communication. These algorithms typically have a quantization level that controls the compression factor. We find that dynamic adaptations of the quantization level can boost compression without sacrificing model quality. First, we introduce a time-adaptive quantization algorithm that increases the quantization level as training progresses. Second, we introduce a client-adaptive quantization algorithm that assigns each individual client the optimal quantization level at every round. Finally, we combine both algorithms into DAdaQuant, the doubly-adaptive quantization algorithm. Our experiments show that DAdaQuant consistently improves client→server compression, outperforming the strongest non-adaptive baselines by up to $2.8\times$.

## 1 Introduction

Edge devices such as smartphones, remote sensors and smart home appliances generate massive amounts of data (Wang et al., 2018b; Cao et al., 2017; Shi & Dustdar, 2016). In recent years, Federated Learning (FL) has emerged as a technique to train models on this data while preserving privacy (McMahan et al., 2017; Li et al., 2018).

In FL, we have a single server that is connected to many clients. Each client stores a local dataset that it does not want to share with the server because of privacy concerns or law enforcement (Voigt & Von dem Bussche, 2017). The server wants to train a model on all local datasets. To this end, it initializes the model and sends it to a random subset of clients. Each client trains the model on its local dataset and sends the trained model back to the server. The server accumulates all trained models into an updated model for the next iteration and repeats the process for several rounds until some termination criterion is met. This procedure enables the server to train a model without accessing any local datasets.

Today's neural network models often have millions or even billions (Brown et al., 2020) of parameters, which makes high communication costs a concern in FL. In fact, Qiu et al. (2020) suggest that communication between clients and server may account for over 70% of energy consumption in FL. Reducing communication in FL is an attractive area of research because it lowers bandwidth requirements, energy consumption and training time.

Communication in FL occurs in two phases: Sending parameters from the server to clients (*downlink*) and sending updated parameters from clients to the server (*uplink*). Uplink bandwidth usually imposes a tighter bottleneck than downlink bandwidth. This has several reasons. For one, the average global mobile upload bandwidth is currently less than one fourth of the download bandwidth (Speedtest). For another, FL downlink communication sends the same parameters to each client. Broadcasting parameters is usually more efficient than the accumulation of parameters from differ-

ent clients that is required for uplink communication (Amiri et al., 2020; Reisizadeh et al., 2019). For these reasons, we seek to compress uplink communication.

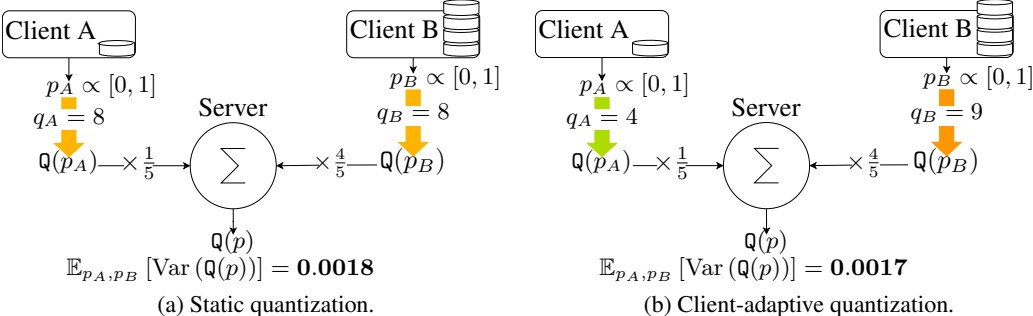

(a) Static quantization.  (b) Client-adaptive quantization.

Figure 1: Static quantization vs. client-adaptive quantization when accumulating parameters $p_A$ and $p_B$. (a): Static quantization uses the same quantization level for $p_A$ and $p_B$. (b) Client-adaptive quantization uses a slightly higher quantization level for $p_B$ because $p_B$ is weighted more heavily. This allows us to use a significantly lower quantization level $q_A$ for $p_A$ while keeping the quantization error measure $\mathrm{E}_{p_A,p_B}[\mathrm{Var}(\mathtt{Q}(p))]$ roughly constant. Since communication is approximately proportional to $q_A + q_B$, client-adaptive quantization communicates less data.

A large class of compression algorithms for FL apply some lossy quantizer $\mathtt{Q}$, optionally followed by a lossless compression stage. $\mathtt{Q}$ usually provides a "quantization level" hyperparameter $q$ to control the coarseness of quantization (e.g. the number of bins for fixed-point quantization). When $q$ is kept constant during training, we speak of *static quantization*. When $q$ changes, we speak of *adaptive quantization*. Adaptive quantization can exploit asymmetries in the FL framework to minimize communication. One such asymmetry lies in FL's training time, where Jhunjhunwala et al. (2021) observed that early training rounds can use a lower $q$ without affecting convergence. Figure 2 illustrates how *time-adaptive quantization* leverages this phenomenon to minimize communication. Another asymmetry lies in FL's client space, because most FL algorithms weight client contributions to the global model proportional to their local dataset sizes. Figure 1 illustrates how *client-adaptive quantization* can minimize the quantization error. Intuitively, FL clients with greater weighting should have a greater commu-

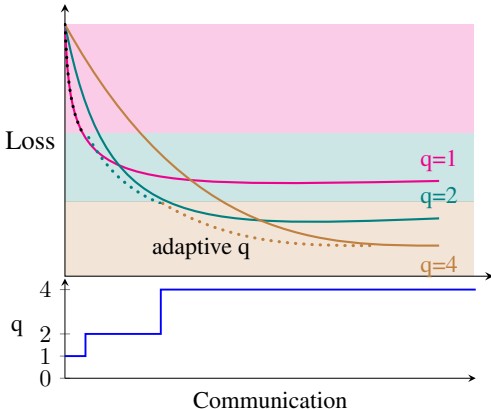

Figure 2: Time-adaptive quantization. A small quantization level (q) decreases the loss with less communication than a large q, but converges to a higher loss. This motivates an adaptive quantization strategy that uses a small q as long as it is beneficial and then switches over to a large q. We generalize this idea into an algorithm that monotonically increases q based on the training loss.

nication budget and our proposed client-adaptive quantization achieves this in a principled way. To this end, we introduce the expected variance of an accumulation of quantized parameters, $\mathbb{E}[\mathrm{Var}(\sum \mathtt{Q}(p))]$, as a measure of the quantization error. Our client-adaptive quantization algorithm then assigns clients minimal quantization levels, subject to a fixed $\mathbb{E}[\mathrm{Var}(\sum \mathtt{Q}(p))]$. This lowers the amount of data communicated from clients to the server, without increasing the quantization error.

DAdaQuant (Doubly Adaptive Quantization) combines time- and client-adaptive quantization with an adaptation of the QSGD fixed-point quantization algorithm to achieve state-of-the-art FL uplink compression. In this paper, we make the following contributions:

- We introduce the concept of client-adaptive quantization and develop algorithms for time- and client-adaptive quantization that are computationally efficient, empirically superior to existing algorithms, and compatible with arbitrary FL quantizers. Our client-adaptive quantization is provably optimal for stochastic fixed-point quantizers.

- We create Federated QSGD as an adaptation of the stochastic fixed-point quantizer QSGD that works with FL. Federated QSGD outperforms all other quantizers, establishing a strong baseline for FL compression with static quantization.
- We combine time- and client-adaptive quantization into DAdaQuant. We demonstrate DAdaQuant's state-of-the-art compression by empirically comparing it against several competitive FL compression algorithms.

## 2 RELATED WORK

FL research has explored several approaches to reduce communication. We identify three general directions.

First, there is a growing interest of investigating FL algorithms that can converge in fewer rounds. FedAvg (McMahan et al., 2017) achieves this with prolonged local training, while FOLB (Nguyen et al., 2020) speeds up convergence through a more principled client sampling. Since communication is proportional to the number of training rounds, these algorithms effectively reduce communication.

Secondly, communication can be reduced by reducing the model size because the model size is proportional to the amount of training communication. PruneFL (Jiang et al., 2019) progressively prunes the model over the course of training, while AFD (Bouacida et al., 2021) only trains sub-models on clients.

Thirdly, it is possible to directly compress FL training communication. FL compression algorithms typically apply techniques like top-k sparsification (Malekijoo et al., 2021; Rothchild et al., 2020) or quantization (Reisizadeh et al., 2019; Shlezinger et al., 2020) to parameter updates, optionally followed by lossless compression. Our work applies to quantization-based compression algorithms. It is partially based on QSGD (Alistarh et al., 2017), which combines lossy fixed-point quantization with a lossless compression algorithm to compress gradients communicated in distributed training. DAdaQuant adapts QSGD into Federated QSGD, which works with Federated Learning. DAdaQuant also draws inspiration from FedPAQ (Reisizadeh et al., 2019), the first FL framework to use lossy compression based on model parameter update quantization. However, FedPAQ does not explore the advantages of additional lossless compression or adaptive quantization. UVeQFed (Shlezinger et al., 2020) is an FL compression algorithm that generalizes scalar quantization to vector quantization and subsequently employs lossless compression with arithmetic coding. Like FedPAQ, UVeQFed also limits itself to a single static quantization level.

Faster convergence, model size reduction and communication compression are orthogonal techniques, so they can be combined for further communication savings. For this paper, we limit the scope of empirical comparisons to quantization-based FL compression algorithms.

For quantization-based compression for model training, prior works have demonstrated that DNNs can be successfully trained in low-precision (Banner et al., 2018; Gupta et al., 2015; Sun et al., 2019). There are also several adaptive quantization algorithms for training neural networks in a non-distributed setting. Shen et al. (2020) use different quantization levels for different parameters of a neural network. FracTrain (Fu et al., 2020) introduced multi-dimensional adaptive quantization by developing time-adaptive quantization and combining it with parameter-adaptive quantization. However, FracTrain uses the current loss to decide on the quantization level. FL generally can only compute local client losses that are too noisy to be practical for FracTrain. AdaQuantFL introduces time-adaptive quantization to FL, but requires the global loss (Jhunjhunwala et al., 2021). To compute the global loss, AdaQuantFL has to communicate with every client each round. We show in Section 4.2 that this quickly becomes impractical as the number of clients grows. DAdaQuant's time-adaptive quantization overcomes this issue without compromising on the underlying FL communication. In addition, to the best of our knowledge, DAdaQuant is the first algorithm to use client-adaptive quantization.

## 3  THE DADAQUANT METHOD

### 3.1  FEDERATED LEARNING

Federated Learning assumes a client-server topology with a set $\mathbb{C} = \{c_i | i \in \{1, 2...N\}\}$ of $N$ clients that are connected to a single server. Each client $c_k$ has a local dataset $D_k$ from the local data distribution $\mathscr{D}_k$. Given a model $M$ with parameters $\boldsymbol{p}$, a loss function $f_{\boldsymbol{p}}(d \in D_k)$ and the local loss $F_k(\boldsymbol{p}) = \frac{1}{|D_k|} \sum_{d \in D_k} f_{\boldsymbol{p}}(d)$, FL seeks to minimize the global loss $G(\boldsymbol{p}) = \sum_{k=1}^{N} \frac{|D_k|}{\sum_l |D_l|} F_k(\boldsymbol{p})$.

### 3.2  FEDERATED AVERAGING (FEDAVG)

DAdaQuant makes only minimal assumptions about the FL algorithm. Crucially, DAdaquant can complement FedAvg (McMahan et al., 2017), which is representative of a large class of FL algorithms.

FedAvg trains the model $M$ over several rounds. In each round $t$, FedAvg sends the model parameters $\boldsymbol{p}_t$ to a random subset $\mathbb{S}_t$ of $K$ clients who then optimize their local objectives $F_k(\boldsymbol{p}_t)$ and send the updated model parameters $\boldsymbol{p}_{t+1}^k$ back to the server. The server accumulates all parameters into the new global model $\boldsymbol{p}_{t+1} = \sum_{k \in \mathbb{S}_t} \frac{|D_k|}{\sum_j |D_j|} \boldsymbol{p}_{t+1}^k$ and starts the next round. Algorithm 1 lists FedAvg in detail. For our experiments, we use the FedProx (Li et al., 2018) adaptation of FedAvg. FedProx improves the convergence of FedAvg by adding the proximal term $\frac{\mu}{2} \|\boldsymbol{p}_{t+1}^k - \boldsymbol{p}_t\|^2$ to the local objective $F_k(\boldsymbol{p}_{t+1}^k)$ in Line 20 of Algorithm 1.

### 3.3  QUANTIZATION WITH FEDERATED QSGD

While DAdaQuant can be applied to any quantizer with a configurable quantization level, it is optimized for fixed-point quantization. We introduce Federated QSGD as a competitive stochastic fixed-point quantizer on top of which DAdaQuant is applied.

In general, stochastic fixed-point quantization uses a quantizer $\mathbb{Q}_q$ with quantization level $q$ that splits $\mathbb{R}_{\geq 0}$ and $\mathbb{R}_{\leq 0}$ into $q$ intervals each. $\mathbb{Q}_q(p)$ then returns the sign of $p$ and $|p|$ stochastically rounded to one of the endpoints of its encompassing interval. $\mathbb{Q}_q(\boldsymbol{p})$ quantizes the vector $\boldsymbol{p}$ elementwise.

We design DAdaQuant's quantization stage based on QSGD, an efficient fixed-point quantizer for state-of-the-art gradient compression. QSGD quantizes a vector $\boldsymbol{p}$ in three steps:

1. Quantize $\boldsymbol{p}$ as $\mathbb{Q}_q(\frac{\boldsymbol{p}}{\|\boldsymbol{p}\|_2})$ into $q$ bins in $[0, 1]$, storing signs and $\|\boldsymbol{p}\|_2$ separately. (*lossy*)
2. Encode the resulting integers with 0 run-length encoding. (*lossless*)
3. Encode the resulting integers with Elias $\omega$ coding. (*lossless*)

QSGD has been designed specifically for quantizing gradients. This makes it not directly applicable to parameter compression. To overcome this limitation, we apply difference coding to uplink compression, first introduced to FL by FedPAQ. Each client $c_k$ applies $\mathbb{Q}_q$ to the *parameter updates* $\boldsymbol{p}_{t+1}^k - \boldsymbol{p}_t$ (cf. Line 21 of Algorithm 1) and sends them to the server. The server keeps track of the previous parameters $\boldsymbol{p}_t$ and accumulates the quantized parameter updates into the new parameters as $\boldsymbol{p}_{t+1} = \boldsymbol{p}_t + \sum_{k \in \mathbb{S}_t} \frac{|D_k|}{\sum_l |D_l|} \mathbb{Q}_q(\boldsymbol{p}_{t+1}^k - \boldsymbol{p}_t)$ (cf. Line 11 of Algorithm 1). We find that QSGD works well with parameter updates, which can be regarded as an accumulation of gradients over several training steps. We call this adaptation of QSGD *Federated QSGD*.

### 3.4  TIME-ADAPTIVE QUANTIZATION

Time-adaptive quantization uses a different quantization level $q_t$ for each round $t$ of FL training. DAdaQuant chooses $q_t$ to minimize communication costs without sacrificing accuracy. To this end, we find that lower quantization levels suffice to initially reduce the loss, while partly trained models require higher quantization levels to further improve (as illustrated in Figure 2). FracTrain is built on similar observations for non-distributed training. Therefore, we design DAdaQuant to mimic FracTrain in monotonically increasing $q_t$ as a function of $t$ and using the training loss to inform increases in $q_t$.

| Round | 1 | 2 | 3 | 4 | 5 |
|---|---|---|---|---|---|
| Client — Samples | | Quantization level | | | |
| A — 1 | | | | 8 | |
| B — 2 | 8 | 8 | 8 | 8 | |
| C — 3 | 8 | | 8 | | 8 |
| D — 4 | | 8 | | | 8 |

(a) Static quantization.

| Round | 1 | 2 | 3 | 4 | 5 |
|---|---|---|---|---|---|
| Client — Samples | | Quantization level | | | |
| A — 1 | | | | 4 | |
| B — 2 | 1 | 2 | 2 | 4 | |
| C — 3 | 1 | | 2 | | 8 |
| D — 4 | | 2 | | | 8 |

(b) Time-adaptive quantization.

| Round | 1 | 2 | 3 | 4 | 5 |
|---|---|---|---|---|---|
| Client — Samples | | Quantization level | | | |
| A — 1 | | | | 6 | |
| B — 2 | 7 | 6 | 7 | 9 | |
| C — 3 | 9 | | 9 | | 7 |
| D — 4 | | 9 | | | 9 |

(c) Client-adaptive quantization.

| Round | 1 | 2 | 3 | 4 | 5 |
|---|---|---|---|---|---|
| Client — Samples | | Quantization level | | | |
| A — 1 | | | | 3 | |
| B — 2 | 1 | 1 | 2 | 5 | |
| C — 3 | 1 | | 2 | | 7 |
| D — 4 | | 1 | | | 9 |

(d) Time-adaptive and client-adaptive quantization.

Figure 3: Exemplary quantization level assignment for 4 FL clients that train over 5 rounds. Each round, two clients get sampled for training.

When $q$ is too low, FL converges prematurely. Like FracTrain, DAdaQuant monitors the FL loss and increases $q$ when it converges. Unlike FracTrain, there is no single centralized loss function to evaluate and unlike AdaQuantFL, we do not assume availability of global training loss $G(\boldsymbol{p}_t)$. Instead, we estimate $G(\boldsymbol{p}_t)$ as the average local loss $\hat{G}_t = \sum_{k \in \mathbb{S}_t} \frac{|D_k|}{\sum_l |D_l|} F_k(\boldsymbol{p}_t)$ where $\mathbb{S}_t$ is the set of clients sampled at round $t$. Since $\mathbb{S}_t$ typically consists of only a small fraction of all clients, $\hat{G}_t$ is a very noisy estimate of $G(\boldsymbol{p}_t)$. This makes it unsuitable for convergence detection. Instead, DAdaQuant tracks a running average loss $\mathring{\hat{G}}_t = \psi \mathring{\hat{G}}_{t-1} + (1-\psi)\hat{G}_t$.

We initialize $q_1 = q_{\min}$ for some $q_{\min} \in \mathbb{N}$. DAdaQuant determines training to converge whenever $\mathring{\hat{G}}_t \geq \mathring{\hat{G}}_{t+1-\phi}$ for some $\phi \in \mathbb{N}$ that specifies the number of rounds across which we compare $\mathring{\hat{G}}$. On convergence, DAdaQuant sets $q_t = 2q_{t-1}$ and keeps the quantization level fixed for at least $\phi$ rounds to enable reductions in $G$ to manifest in $\mathring{\hat{G}}$. Eventually, the training loss converges regardless of the quantization level. To avoid unconstrained quantization increases on convergence, we limit the quantization level to $q_{\max}$.

The following equation summarizes DAdaQuant's time-adaptive quantization:

$$q_t \longleftarrow \begin{cases} q_{\min} & t = 0 \\ 2q_{t-1} & t > 0 \text{ and } \mathring{\hat{G}}_{t-1} \geq \mathring{\hat{G}}_{t-\phi} \text{ and } t > \phi \text{ and } 2q_{t-1} < q_{\max} \text{ and } q_{t-1} = q_{t-\phi} \\ q_{t-1} & \text{else} \end{cases}$$

## 3.5 CLIENT-ADAPTIVE QUANTIZATION

FL algorithms typically accumulate each parameter $p_i$ over all clients into a weighted average $p = \sum_{i=1}^{K} w_i p_i$ (see Algorithm 1). Quantized FL accumulates quantized parameters $\mathsf{Q}_q(p) = \sum_{i=1}^{K} w_i \mathsf{Q}_q(p_i)$ where $q$ is the quantization level. We define the quantization error $e_p^q = |p - \mathsf{Q}_q(p)|$.

We observe in our experiments that communication cost per client is roughly a linear function of Federated QSGD's quantization level $q$. This means that the communication cost per round is proportional to $Q = Kq$. We call $Q$ the communication budget and use it as a proxy measure of communication cost.

Client-adaptive quantization dynamically adjusts the quantization level of each client. This means that even within a single round, each client $c_k$ can be assigned a different quantization level $q_k$. The previous definitions then generalize to $Q = \sum_{k=1}^{K} q_k$ and $\mathsf{Q}_{q_1 \ldots q_K}(p) = \sum_{i=1}^{K} w_i \mathsf{Q}_{q_i}(p_i)$ and $e_p^{q_1 \ldots q_K} = |p - \mathsf{Q}_{q_1 \ldots q_K}(p)|$.

Prior convergence results for distributed training and FL rely on an upper bound $b$ on $\text{Var}(\mathsf{Q}_{q_1 \ldots q_K}(p))$ that determines the convergence speed Li et al. (2017); Horváth et al. (2019); Reisizadeh et al. (2019). This makes $\text{V}(\mathsf{Q}_{q_1 \ldots q_K}(p))$ a natural measure to optimize for when choosing $q_k$.

We optimize for the closely related measure $\mathbb{E}_{p_1 \ldots p_K}[\text{Var}(\mathbb{Q}_{q_1 \ldots q_K}(p))]$ that replaces the upper bound with an expectation over parameters $p_1 \ldots p_K$. Heuristically, we expect an this averaged measure to provide a better estimate of practically observed quantization errors than an upper bound. For a stochastic, unbiased fixed-point compressor like Federated QSGD, $\mathbb{E}_{p_1 \ldots -p_K}[\text{Var}(\mathbb{Q}_{q_1 \ldots q_K}(p))]$ equals $\mathbb{E}_{p_1 \ldots p_K}[\text{Var}(e_p^q)]$ and can be evaluated analytically.

We devise an algorithm that chooses $q_k$ to minimize $Q$ subject to $\mathbb{E}_{p_1 \ldots p_K}[\text{Var}(e_p^{q_1 \ldots q_K})] = \mathbb{E}_{p_1 \ldots p_K}[\text{Var}(e_p^q)]$ for a given $q$. Thus, our algorithm effectively minimizes communication costs while maintaining a quantization error similar to static quantization. Theorem 1 provides us with an analytical formula for quantization levels $q_1 \ldots q_K$.

**Theorem 1.** *Given parameters $p_1 \ldots p_k \sim \mathcal{U}[-t, t]$ and quantization level $q$, $\min_{q_1 \ldots q_K} \sum_{i=1}^{K} q_i$ subject to $\mathbb{E}_{p_1 \ldots p_K}[\text{Var}(e_p^{q_1 \ldots q_K})] = \mathbb{E}_{p_1 \ldots p_K}[\text{Var}(e_p^q)]$ is minimized by $q_i = \sqrt{\frac{a}{b}} \times w_i^{2/3}$ where $a = \sum_{j=1}^{K} w_j^{2/3}$ and $b = \sum_{j=1}^{K} \frac{w_j^2}{q^2}$.*

DAdaQuant applies Theorem 1 to lower communication costs while maintaining the same loss as static quantization does with a fixed $q$. To ensure that quantization levels are natural numbers, DAdaQuant approximates the optimal real-valued solution as $q_i = \max(1, \text{round}(\sqrt{\frac{a}{b}} \times w_i^{2/3}))$. Appendix B gives a detailed proof of Theorem 1. To the best of our knowledge, DAdaQuant is the first algorithm to use client-adaptive quantization.

---

**Algorithm 1:** The FedAvg and DAdaQuant algorithms. The uncolored lines list FedAvg. Adding the colored lines creates DAdaQuant. ■ — quantization, ■ — client-adaptive quantization, ■ — time-adaptive quantization.

1 **Function** RunServer()
2     Initialize $w_i = \frac{|D_i|}{\sum_j |D_j|}$ for all $i \in [1, \ldots, N]$;
3     **for** $t = 0, \ldots, T - 1$ **do**
4         Choose $\mathbb{S}_t \subset \mathbb{C}$ with $|\mathbb{S}_t| = K$, including each $c_k \in \mathbb{C}$ with uniform probability;
5         $q_t \longleftarrow \begin{cases} q_{\min} & t = 0 \\ 2q_{t-1} & t > 0 \text{ and } \hat{\hat{G}}_{t-1} \geq \hat{\hat{G}}_{t-\phi} \text{ and } t > \phi \text{ and } q_t \leq q_{\max} \text{ and } q_{t-1} = q_{t-\phi} \\ q_{t-1} & \text{else} \end{cases}$ ;
6         **for** $c_k \in \mathbb{S}_t$ **do** in parallel
7             $q_t^k \longleftarrow \sqrt{\sum_{j=1}^{K} w_j^{2/3} / \sum_{j=1}^{K} \frac{w_j^2}{q^2}}$ ;
8             $Send(c_k, \boldsymbol{p}_t, q_t^k)$;
9             $Receive(c_k, \boldsymbol{p}_{t+1}^k, \hat{G}_t^k)$;
10        **end**
11        $\boldsymbol{p}_{t+1} \longleftarrow \sum_{k \in \mathbb{S}_t} w_k \boldsymbol{p}_{t+1}^k$;
12        $\hat{G}_t \longleftarrow \sum_{k \in \mathbb{S}_t} w_k \hat{G}_t^k$ ;
13        $\hat{\hat{G}}_t \longleftarrow \begin{cases} \hat{G}_0 & t = 0 \\ \psi \hat{\hat{G}}_{t-1} + (1 - \psi)\hat{G}_t & \text{else} \end{cases}$ ;
14    **end**
15 **end**
16 **Function** RunClient($c_k$)
17    **while** True **do**
18        $Receive(\text{Server}, \boldsymbol{p}_t, q_t^k)$;
19        $\hat{G}_t^k \longleftarrow F_k(\boldsymbol{p}_t)$ ;
20        $\boldsymbol{p}_{t+1}^k \longleftarrow F_k(\boldsymbol{p}_{t+1}^k)$ trained with SGD for $E$ epochs with learning rate $\eta$;
21        $Send(\text{Server}, \mathbb{Q}_{q_t^k}(\boldsymbol{p}_{t+1}^k), \hat{G}_t^k)$;
22    **end**
23 **end**

---

### 3.6 DOUBLY-ADAPTIVE QUANTIZATION (DADAQUANT)

DAdaQuant combines the time-adaptive and client-adaptive quantization algorithms described in the previous sections. At each round $t$, time-adaptive quantization determines a preliminary quantization level $q_t$. Client-adaptive quantization then finds the client quantization levels $q_t^k, k \in \{1, \ldots, K\}$ that minimize $\sum_{i=1}^{K} q_i$ subject to $\mathbb{E}_{p_1 \ldots p_K}[\text{Var}(e_p^{q_1 \cdots q_K})] = \mathbb{E}_{p_1 \ldots p_K}[\text{Var}(e_p^q)]$. Algorithm 1 lists DAdaQuant in detail. Figure 3 gives an example of how our time-adaptive, client-adaptive and doubly-adaptive quantization algorithms set quantization levels.

Reisizadeh et al. (2019) prove the convergence of FL with quantization for convex and non-convex cases as long as the quantizer Q is (1) unbiased and (2) has a bounded variance. These convergence results extend to DAdaQuant when combined with any quantizer that satisfies (1) and (2) for DAdaQuant's minimum quantization level $q = 1$. Crucially, this includes Federated QSGD.

We highlight DAdaQuant's low overhead and general applicability. The computational overhead is dominated by an additional evaluation epoch per round per client to compute $\hat{\hat{G}}_t$, which is negligible when training for many epochs per round. In our experiments, we observe computational overheads of $\approx 1\%$ (see Appendix A.2). DAdaQuant can compliment any FL algorithm that trains models over several rounds and accumulates a weighted average of client parameters. Most FL algorithms, including FedAvg, follow this design.

## 4 EXPERIMENTS

### 4.1 EXPERIMENTAL DETAILS

**Evaluation** We use DAdaQuant with Federated QSGD to train different models with FedProx on different datasets for a fixed number of rounds. We monitor the test loss and accuracy at fixed intervals and measure uplink communication at every round across all devices.

**Models & datasets** We select a broad and diverse set of five models and datasets to demonstrate the general applicability of DAdaQuant. To this end, we use DAdaQuant to train a linear model, CNNs and LSTMs of varying complexity on a federated synthetic dataset (Synthetic), as well as two federated image datasets (FEMNIST and CelebA) and two federated natural language datasets (Sent140 and Shakespeare) from the LEAF (Caldas et al., 2018) project for standardized FL research. We refer to Appendix A.1 for more information on the models, datasets, training objectives and implementation.

**System heterogeneity** In practice, FL has to cope with clients that have different compute capabilities. We follow Li et al. (2018) and simulate this *system heterogeneity* by randomly reducing the number of epochs to $E'$ for a random subset $\mathbb{S}'_t \subset \mathbb{S}_t$ of clients at each round $t$, where $E'$ is sampled from $[1, \ldots, E]$ and $|\mathbb{S}'_t| = 0.9K$.

**Baselines** We compare DAdaQuant against competing quantization-based algorithms for FL parameter compression, namely Federated QSGD, FedPAQ (Reisizadeh et al., 2019), GZip with fixed-point quantization (FxPQ + GZip), UVeQFed (Shlezinger et al., 2020) and FP8. Federated QSGD (see section 3.3) is our most important baseline because it outperforms the other algorithms. FedPAQ only applies fixed-point quantization, which is equivalent to Federated QSGD without lossless compression. Similarly, FxPQ + GZip is equivalent to Federated QSGD with Gzip for its lossless compression stages. UVeQFed generalizes scalar quantization to vector quantization, followed by arithmetic coding. We apply UVeQFed with the optimal hyperparameters reported by its authors. FP8 (Wang et al., 2018a) is a floating-point quantizer that uses an 8-bit floating-point format designed for storing neural network gradients. We also evaluate all experiments without compression to establish an accuracy benchmark.

**Hyperparameters** With the exception of CelebA, all our datasets and models are also used by Li et al.. We therefore adopt most of the hyperparameters from Li et al. and use LEAF's hyperparameters for CelebA Caldas et al. (2018). For all experiments, we sample 10 clients each round. We train Synthetic, FEMNIST and CelebA for 500 rounds each. We train Sent140 for 1000 rounds due to slow convergence and Shakespeare for 50 rounds due to rapid convergence. We use batch size 10, learning rates 0.01, 0.003, 0.3, 0.8, 0.1 and $\mu$s (FedProx's proximal term coefficient) 1, 1, 1, 0.001, 0

| | Synthetic | | FEMNIST | | Sent140 | |
|---|---|---|---|---|---|---|
| Uncompressed | $78.3 \pm 0.3$ | 12.2 MB | $77.7 \pm 0.4$ | 132.1 GB | $69.7 \pm 0.5$ | 43.9 GB |
| Federated QSGD | $-0.1 \pm 0.1$ | $17\times$ | $+0.7 \pm 0.5$ | $2809\times$ | $-0.0 \pm 0.5$ | $90\times$ |
| FP8 | $\mathbf{+0.1 \pm 0.4}$ | $4.0\times\,(0.23\times)$ | $-0.1 \pm 0.4$ | $4.0\times\,(0.00\times)$ | $-0.2 \pm 0.5$ | $4.0\times\,(0.04\times)$ |
| FedPAQ (FxPQ) | $-0.1 \pm 0.1$ | $6.4\times\,(0.37\times)$ | $+0.7 \pm 0.5$ | $11\times\,(0.00\times)$ | $-0.0 \pm 0.5$ | $4.0\times\,(0.04\times)$ |
| FxPQ + GZip | $-0.1 \pm 0.1$ | $14\times\,(0.82\times)$ | $+0.6 \pm 0.2$ | $1557\times\,(0.55\times)$ | $-0.0 \pm 0.6$ | $71\times\,(0.79\times)$ |
| UVeQFed | $-0.5 \pm 0.2$ | $0.6\times\,(0.03\times)$ | $-2.8 \pm 0.5$ | $12\times\,(0.00\times)$ | $+0.0 \pm 0.2$ | $15\times\,(0.16\times)$ |
| DAdaQuant | $-0.2 \pm 0.4$ | $\mathbf{48\times\,(2.81\times)}$ | $+0.7 \pm 0.1$ | $\mathbf{4772\times\,(1.70\times)}$ | $-0.1 \pm 0.4$ | $\mathbf{108\times\,(1.19\times)}$ |
| DAdaQuant$_{time}$ | $-0.1 \pm 0.5$ | $37\times\,(2.16\times)$ | $\mathbf{+0.8 \pm 0.2}$ | $4518\times\,(1.61\times)$ | $-0.1 \pm 0.6$ | $93\times\,(1.03\times)$ |
| DAdaQuant$_{clients}$ | $+0.0 \pm 0.3$ | $26\times\,(1.51\times)$ | $+0.7 \pm 0.4$ | $3017\times\,(1.07\times)$ | $\mathbf{+0.1 \pm 0.6}$ | $105\times\,(1.16\times)$ |

| | Shakespeare | | Celeba | |
|---|---|---|---|---|
| Uncompressed | $\mathbf{49.9 \pm 0.3}$ | 267.0 MB | $90.4 \pm 0.0$ | 12.6 GB |
| Federated QSGD | $-0.5 \pm 0.6$ | $9.5\times$ | $-0.1 \pm 0.1$ | $648\times$ |
| FP8 | $-0.2 \pm 0.4$ | $4.0\times\,(0.42\times)$ | $\mathbf{+0.0 \pm 0.1}$ | $4.0\times\,(0.01\times)$ |
| FedPAQ (FxPQ) | $-0.5 \pm 0.6$ | $3.2\times\,(0.34\times)$ | $-0.1 \pm 0.1$ | $6.4\times\,(0.01\times)$ |
| FxPQ + GZip | $-0.5 \pm 0.6$ | $9.3\times\,(0.97\times)$ | $-0.1 \pm 0.2$ | $494\times\,(0.76\times)$ |
| UVeQFed | $-0.0 \pm 0.4$ | $7.9\times\,(0.83\times)$ | $-0.4 \pm 0.3$ | $31\times\,(0.05\times)$ |
| DAdaQuant | $-0.6 \pm 0.5$ | $\mathbf{21\times\,(2.21\times)}$ | $-0.1 \pm 0.1$ | $\mathbf{775\times\,(1.20\times)}$ |
| DAdaQuant$_{time}$ | $-0.5 \pm 0.5$ | $12\times\,(1.29\times)$ | $-0.1 \pm 0.2$ | $716\times\,(1.10\times)$ |
| DAdaQuant$_{clients}$ | $-0.4 \pm 0.5$ | $16\times\,(1.67\times)$ | $-0.1 \pm 0.0$ | $700\times\,(1.08\times)$ |

Table 1: Top-1 test accuracies and total client→server communication of all baselines, DAdaQuant, DAdaQuant$_{time}$ and DAdaQuant$_{clients}$. Entry $x \pm y \quad p\times\,(q\times)$ denotes an accuracy difference of x% w.r.t. the uncompressed accuracy with a standard deviation of y%, a compression factor of $p$ w.r.t. the uncompressed communication and a compression factor of $q$ w.r.t. Federated QSGD.

for Synthetic, FEMNIST, Sent140, Shakespeare, CelebA respectively. We randomly split the local datasets into 80% training set and 20% test set.

To select the quantization level $q$ for static quantization with Federated QSGD, FedPAQ and FxPQ + GZip, we run a gridsearch over $q = 1, 2, 4, 8, \ldots$ and choose for each dataset the lowest $q$ for which Federated QSGD exceeds uncompressed training in accuracy. We set UVeQFed's "coding rate" hyperparameter $R = 4$, which is the lowest value for which UVeQFed achieves negligible accuracy differences compared to uncompressed training. We set the remaining hyperparameters of UVeQFed to the optimal values reported by its authors. Appendix A.4 shows further experiments that compare against UVeQFed with $R$ chosen to maximize its compression factor.

For DAdaQuant's time-adaptive quantization, we set $\psi$ to 0.9, $\phi$ to $1/10^{th}$ of the number of rounds and $q_{max}$ to the quantization level $q$ for each experiment. For Synthetic and FEMNIST, we set $q_{min}$ to 1. We find that Sent140, Shakespeare and CelebA require a high quantization level to achieve top accuracies and/or converge in few rounds. This prevents time-adaptive quantization from increasing the quantization level quickly enough, resulting in prolonged low-precision training that hurts model performance. To counter this effect, we set $q_{min}$ to $q_{max}/2$. This effectively results in binary time-adaptive quantization with an initial low-precision phase with $q = q_{max}/2$, followed by a high-precision phase with $q = q_{max}$.

## 4.2 RESULTS

We repeat the main experiments three times and report average results and their standard deviation (where applicable). Table 1 shows the highest accuracy and total communication for each experiment. Figure 4 plots the maximum accuracy achieved for any given amount of communication.

**Baselines** Table 1 shows that the accuracy of most experiments lies within the margin of error of the uncompressed experiments. This reiterates the viability of quantization-based compression algorithms for communication reduction in FL. For all experiments, Federated QSGD achieves a significantly higher compression factor than the other baselines. The authors of FedPAQ and UVe-QFed also compare their methods against QSGD and report them as superior. However, FedPAQ is compared against "unfederated" QSGD that communicates gradients after each local training step and UVeQFed is compared against QSGD without its lossless compression stages.

**Time-adaptive quantization** The purely time-adaptive version of DAdaQuant, DAdaQuant$_{time}$, universally outperforms Federated QSGD and the other baselines in Table 1, achieving comparable accuracies while lowering communication costs. DAdaQuant$_{time}$ performs particularly well on Syn-

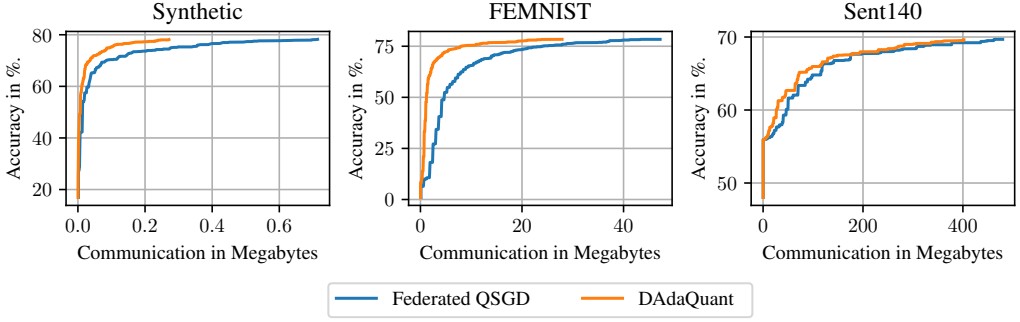

Figure 4: Communication-accuracy trade-off curves for training on FEMNIST with Federated QSGD and DAdaQuant. We plot the average highest accuracies achieved up to any given amount of client→server communication. Appendix A.3 shows curves for all datasets, as well as DAdaQuant$_{time}$ and DAdaQuant$_{clients}$, with similar results.

thetic and FEMNIST, where it starts from the lowest possible quantization level $q = 1$. However, binary time-adaptive quantization still measurably improves over QSGD for Sent140, Shakespeare and Celeba.

Figure 8 in Appendix A.5 provides empirical evidence that AdaQuantFL's communication scales linearly with the number of clients. As a result, AdaQuantFL is prohibitively expensive for datasets with thousands of clients such as Celeba and Sent140. DAdaQuant does not face this problem because its communication is unaffected by the number of clients.

**Client-adaptive quantization** The purely time-adaptive version of DAdaQuant, DAdaQuant$_{clients}$, also universally outperforms Federated QSGD and the other baselines in Table 1, achieving similar accuracies while lowering communication costs. Unsurprisingly, the performance of DAdaQuant$_{clients}$ is correlated with the coefficient of variation $c_v = \frac{\sigma}{\mu}$ of the numbers of samples in the local datasets with mean $\mu$ and standard deviation $\sigma$: Synthetic ($c_v = 3.3$) and Shakespeare ($c_v = 1.7$) achieve significantly higher compression factors than Sent140 ($c_v = 0.3$), FEMNIST ($c_v = 0.4$) and Celeba ($c_v = 0.3$).

**DAdaQuant** DAdaQuant outperforms DAdaQuant$_{time}$ and DAdaQuant$_{clients}$ in communication while achieving similar accuracies. The compression factors of DAdaQuant are roughly multiplicative in those of DAdaQuant$_{clients}$ and DAdaQuant$_{time}$. This demonstrates that we can effectively combine time- and client-adaptive quantization for maximal communication savings. Figure 4 shows that DAdaQuant achieves a higher accuracy than the strongest baseline, Federated QSGD, for any fixed amount of client→server communication.

## 5    CONCLUSION

We introduced DAdaQuant as a computationally efficient and robust algorithm to boost the performance of quantization-based FL compression algorithms. We showed intuitively and mathematically how DAdaQuant's dynamic adjustment of the quantization level across time and clients minimize client→server communication while maintaining convergence speed. Our experiments establish DAdaQuant as nearly universally superior over static quantizers, achieving state-of-the-art compression factors when applied to Federated QSGD. The communication savings of DAdaQuant effectively lower FL bandwidth usage, energy consumption and training time. Future work may apply and adapt DAdaQuant to new quantizers, further pushing the state of the art in FL uplink compression.

## 6    REPRODUCIBILITY STATEMENT

Our submission includes a repository with the source code for DAdaQuant and for the experiments presented in this paper. All the datasets used in our experiments are publicly available. Any post-processing steps of the datasets are described in Appendix A.1. To facilitate the reproduction of

our results, we have bundled all our source code, dependencies and datasets into a Docker image. The repository submitted with this paper contains instructions on how to use this Docker image and reproduce all plots and tables in this paper.

## 7 ETHICS STATEMENT

FL trains models on private client datasets in a privacy-preserving manner. However, FL does not completely eliminate privacy concerns, because the transmitted model updates and the learned model parameters may expose the private client data from which they are derived. Our work does not directly target privacy concerns in FL. With that said, it is worth noting that DAdaQuant does not expose any client data that is not already exposed through standard FL training algorithms. In fact, DAdaQuant reduces the amount of exposed data through lossy compression of the model updates. We therefore believe that DAdaQuant is free of ethical complications.

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

# A  ADDITIONAL SIMULATION DETAILS AND EXPERIMENTS

## A.1  ADDITIONAL SIMULATION DETAILS

Here, we give detailed information on the models, datasets, training objectives and implementation that we use for our experiments. We set the five following FL tasks:

- Multinomial logistic regression (MLR) on a synthetic dataset called Synthetic that contains vectors in $\mathbb{R}^{60}$ with a label of one out of 10 classes. We use the synthetic dataset generator in Li et al. (2018) to generate synthetic datasets. The generator samples Synthetic's local datasets and labels from MLR models with randomly initialized parameters. For this purpose, parameters $\alpha$ and $\beta$ control different kinds of data heterogeneity. $\alpha$ controls the variation in the local models from which the local dataset labels are generated. $\beta$ controls the variation in the local dataset samples. We set $\alpha = 1$ and $\beta = 1$ to simulate an FL setting with both kinds of data heterogeneity. This makes Synthetic a useful testbed for FL.
- Character classification into 62 classes of handwritten characters from the FEMNIST dataset using a CNN. FEMNIST groups samples from the same author into the same local dataset.
- Smile detection in facial images from the CelebA dataset using a CNN. CelebA groups samples of the same person into the same local dataset. We note that LEAF's CNN for CelebA uses BatchNorm layers. We replace them with LayerNorm layers because they are more amenable to quantization. This change does not affect the final accuracy.
- Binary sentiment analysis of tweets from the Sent140 dataset using an LSTM. Sent140 groups tweets from the same user into the same local dataset. The majority of local datasets in the raw Sent140 dataset only have a single sample. This impedes FL convergence. Therefore, we filter Sent140 to clients with at least 10 samples (i.e. one complete batch). Caldas et al. (2018); Li et al. (2018) similarly filter Sent140 for their FL experiments.
- Next character prediction on text snippets from the Shakespeare dataset of Shakespeare's collected plays using an LSTM. Shakespeare groups lines from the same character into the same local dataset.

Table 2 provides statistics of our models and datasets.

For our experiments in Figure 8, AdaQuantFL requires a hyperparameter $s$ that determines the initial quantization level. We set $s$ to 2, the optimal value reported by the authors of AdaQuantFL. The remaining hyperparameters are identical to those used for the Synthetic dataset experiments in Table 1.

We implement the models with PyTorch (Paszke et al., 2019) and use Flower (Beutel et al., 2020) to simulate the FL server and clients.

| Dataset | Model | Parameters | Clients | Samples | Samples per client | | | |
|---|---|---|---|---|---|---|---|---|
| | | | | | mean | min | max | stddev |
| Synthetic | MLR | 610 | 30 | 9,600 | 320.0 | 45 | 5,953 | 1051.6 |
| FEMNIST | 2-layer CNN | $6.6 \times 10^6$ | 3,500 | 785,582 | 224.1 | 19 | 584 | 87.8 |
| CelebA | 4-layer CNN | $6.3 \times 10^5$ | 9,343 | 200,288 | 21.4 | 5 | 35 | 7.6 |
| Sent140 | 2-layer LSTM | $1.1 \times 10^6$ | 21,876 | 430,707 | 51.1 | 10 | 549 | 17.1 |
| Shakespeare | 2-layer LSTM | $1.3 \times 10^5$ | 1,129 | 4,226,158 | 3743 | 3 | 66,903 | 6212 |

Table 2: Statistics of the models and datasets used for evaluation. MLR stands for "Multinomial Logistic Regression".

## A.2  COMPUTATIONAL OVERHEAD OF DADAQUANT

| Training | DAdaQuant$_{time}$ | DAdaQuant$_{clients}$ | Federated QSGD | Total overhead |
|---|---|---|---|---|
| 36 s | <1 ms (0.00%) | 0.17 s (0.47%) | 0.24 s (0.67%) | 0.41 s (1.14%) |

Table 3: Execution time measurements for different stages of a FL training round on FEMNIST with DAdaQuant. Each entry contains the execution time in seconds and as a fraction of the normal training time. The total overhead of DAdaQuant, including Federated QSGD, is $\approx 1\%$.

### A.3 COMPLETE COMMUNICATION-ACCURACY TRADE-OFF CURVES

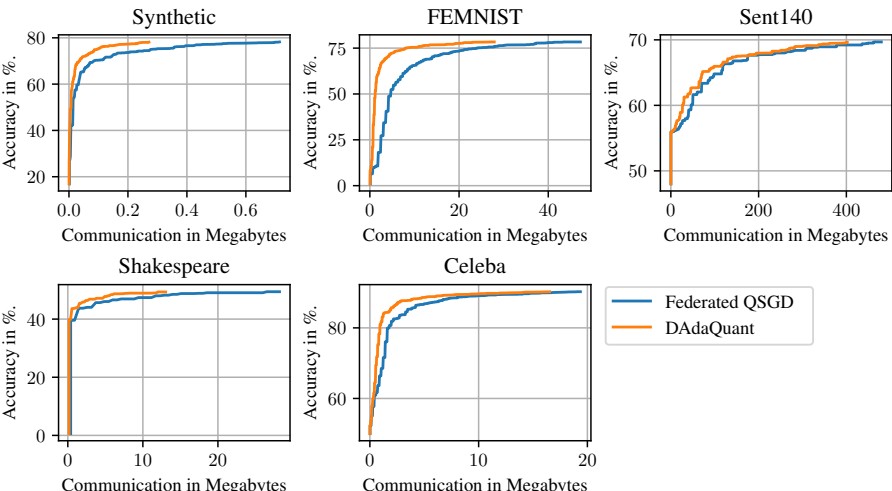

Figure 5: Communication-accuracy trade-off curves for Federated QSGD and DAdaQuant. We plot the average highest accuracies achieved up to any given amount of communication.

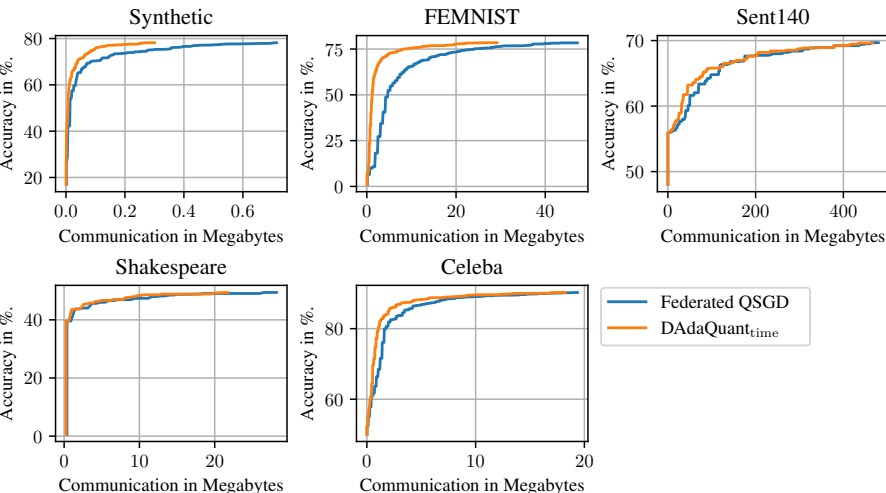

Figure 6: Communication-accuracy trade-off curves for Federated QSGD and DAdaQuant$_{\text{time}}$. We plot the average highest accuracies achieved up to any given amount of communication.

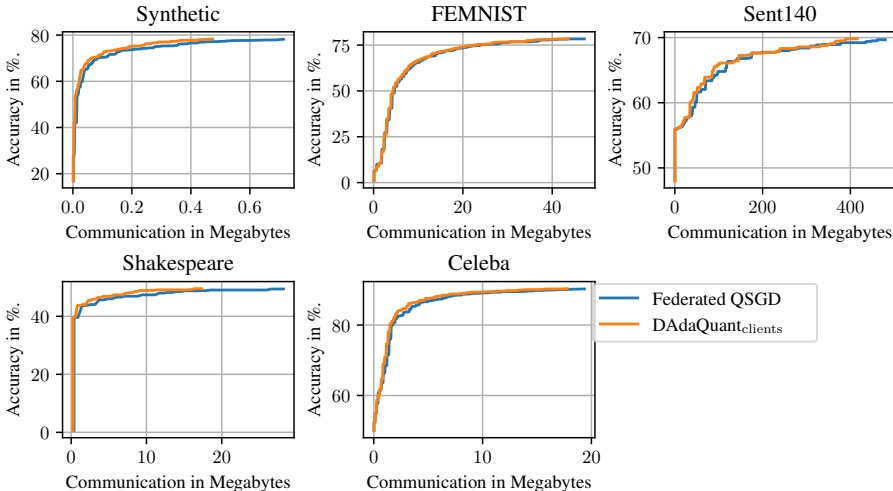

Figure 7: Communication-accuracy trade-off curves for Federated QSGD and DAdaQuant$_{\text{clients}}$. We plot the average highest accuracies achieved up to any given amount of communication.

## A.4 Additional UVeQFed experiments

To demonstrate that the choice of UVeQFed's "coding rate" hyperparameter $R$ does not affect our findings on the superior compression factors of DAdaQuant, we re-evaluate UVeQFed with $R = 1$, which maximizes UVeQFed's compression factor. Our results in Table 4 show that with the exception of Shakespeare, DAdaQuant still achieves considerably higher compression factors than UVeQFed.

| | Synthetic | FEMNIST | Sent140 | Shakespeare | Celeba |
|---|---|---|---|---|---|
| **Uncompressed** | 12.2 MB | 132.1 GB | 43.9 GB | 267.0 MB | 12.6 GB |
| **QSGD** | 17× | 2809× | 90× | 9.5× | 648× |
| **UVeQFed (R=4)** | 0.6× (0.03 ✳) | 12× (0.00 ✳) | 15× (0.16 ✳) | 7.9× (0.83 ✳) | 31× (0.05 ✳) |
| **UVeQFed (R=1)** | 13× (0.77 ✳) | 34× (0.01 ✳) | 41× (0.45 ✳) | **21× (2.22 ✳)** | 93× (0.14 ✳) |
| **DAdaQuant** | **48× (2.81 ✳)** | **4772× (1.70 ✳)** | **108× (1.19 ✳)** | 21× (2.21 ✳) | **775× (1.20 ✳)** |

Table 4: Comparison of the compression factors of DAdaQuant, UVeQFed with $R = 4$ (default value used for our experiments in Table 1) and UVeQFed with $R = 1$ (maximizes UVeQFed's compression factor). Entry $p \times (q ✳)$ denotes a compression factor of $p$ w.r.t. the uncompressed communication and a compression factor of $q$ w.r.t. Federated QSGD.

## A.5 ADDITIONAL ADAQUANTFL EXPERIMENTS

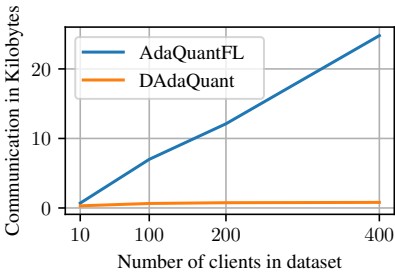

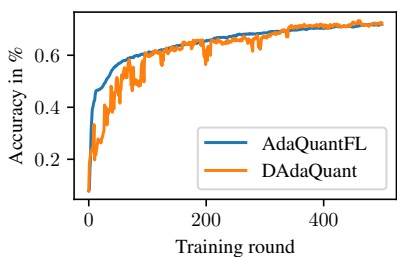

(a) Comparison of the per-round-communication for AdaQuantFL and DAdaQuant. We plot the average client→server communication per round that is required to train an MLR model on synthetic datasets with 10, 100, 200 and 400 clients. AdaQuantFL's communication increases linearly with the number of clients because it trains the model on all clients at each round. In contrast, DAdaQuant's communication does not change with the number of clients.

(b) Comparison of the convergence speed for AdaQuantFL and DAdaQuant. We plot the test accuracy while training on a synthetic dataset with 100 clients. Although AdaQuantFL has full client participation each round, it converges only slightly faster than DAdaQuant and achieves a similar top accuracy. This means that AdaQuantFL's linear increase in communication is *not* offset by a proportional reduction in training rounds.

Figure 8: Scalability of AdaQuantFL vs. DAdaQuant.

In principle, AdaQuantFL could be adapted to work with partial client participation by computing an estimate of the global loss from the sampled subset of clients. While a full evaluation of this approach is out of the scope of this paper, we conduct a brief feasibility study on FEMNIST. Concretely, we find that a single run of AdaQuantFL with partial client participation on FEMNIST achieved an accuracy of 78.7%, with a total client→server communication of 50.5 MB. In contrast, the same run with DAdaQuant$_{\text{time}}$ similarly achieved an accuracy of 78.4%, while lowering the total client→server communication to 27.5 MB.

## B PROOFS

**Lemma 1.** *Take arbitrary quantization level* $q_i \in \mathbb{N}$ *and parameter* $p_i \in [-t, t]$. *Then,* $\mathbb{Q}_{q_i}(p_i)$ *is an unbiased estimator of* $p_i$.

*Proof.* Let $s_i = \frac{t}{q_i}$, $b_i = \text{rem}(p_i, s_i)$ and $u_i = s_i - b_i$. Then, we have

$$
\mathrm{E}\left[\mathbb{Q}_{q_i}(p_i) - p_i\right]
$$
$$
= \frac{u_i}{s_i}(p_i - b_i) + \frac{b_i}{s_i}(p_i + u_i) \qquad \text{see Figure 9}
$$
$$
= p_i \qquad \qquad \square
$$

**Lemma 2.** *For arbitrary* $t > 0$ *and parameter* $p_i \in [-t, t]$, *let* $s_i = \frac{t}{q_i}$, $b_i = \text{rem}(p_i, s_i)$ *and* $u_i = s_i - b_i$. *Then,* $\text{Var}\left(\mathbb{Q}_{q_i}(p_i)\right) = u_i b_i$.

*Proof.*

$$
\text{Var}\left(\mathbb{Q}_{q_i}(p_i)\right)
$$
$$
= \mathrm{E}\left[\left(\mathbb{Q}_{q_i}(p_i) - \mathrm{E}\left[\mathbb{Q}_{q_i}(p_i)\right]\right)^2\right]
$$
$$
= \mathrm{E}\left[\left(\mathbb{Q}_{q_i}(p_i) - p_i\right)^2\right] \qquad \text{see Lemma 1}
$$
$$
= \frac{b_i}{s_i}u_i^2 + \frac{u_i}{s_i}b_i^2 \qquad \text{see Figure 9}
$$

$$= \frac{u_i b_i}{s_i} (u_i + b_i)$$

$$= u_i b_i \qquad \qquad \square$$

$$P(cs_i) = \tfrac{u_i}{s_i} \qquad\qquad\qquad P((c+1)s_i) = \tfrac{b_i}{s_i}$$

$$cs_i \qquad\qquad\qquad p_i \quad (c+1)s_i$$

Figure 9: Illustration of the Bernoulli random variable $Q_{q_i}(p_i)$. $s_i$ is the length of the quantization interval. $p_i$ is rounded up to $(c+1)s_i$ with a probability proportional to its distance from $cs_i$.

**Lemma 3.** *Assume that parameters $p_1 \ldots p_K$ are sampled from $\mathcal{U}[-t, t]$ for arbitrary $t > 0$. Then,* $\mathbb{E}_{p_1 \ldots p_K}[\mathrm{Var}(e_p^{q_1 \ldots q_K})] = \frac{t^2}{6} \sum_{i=1}^{K} \frac{w_i^2}{q_i^2}.$

*Proof.*

$$\mathbb{E}_{p_1 \ldots p_K}[\mathrm{Var}(e_p)]$$

$$= \frac{1}{2t} \int_{-t}^{t} \frac{1}{2t} \int_{-t}^{t} \cdots \frac{1}{2t} \int_{-t}^{t} \mathrm{Var}\left( \sum_{i=1}^{K} w_i Q_{q_i}(p_i) - p \right) dp_1 dp_2 \ldots dp_K$$

$$= \frac{1}{t} \int_{0}^{t} \frac{1}{t} \int_{0}^{t} \cdots \frac{1}{t} \int_{0}^{t} \mathrm{Var}\left( \sum_{i=1}^{K} w_i Q_{q_i}(p_i) - p \right) dp_1 dp_2 \ldots dp_K \qquad \text{symmetry of } Q_{q_i}(p_i) \\ \text{w.r.t. negation}$$

$$= \frac{1}{t^n} \int_{0}^{t} \int_{0}^{t} \cdots \int_{0}^{t} \sum_{i=1}^{K} w_i^2 \mathrm{Var}\left( Q_{q_i}(p_i) \right) dp_1 dp_2 \ldots dp_K \qquad \text{mutual independence of } Q_{q_i}(p_i) \ \forall i$$

$$= \frac{1}{t^n} \sum_{i=1}^{K} \int_{0}^{t} \int_{0}^{t} \cdots \int_{0}^{t} w_i^2 \mathrm{Var}\left( Q_{q_i}(p_i) \right) dp_1 dp_2 \ldots dp_K \qquad \text{exchangeability of finite sums and integrals}$$

$$= \frac{1}{t^n} \sum_{i=1}^{K} t^{n-1} \int_{0}^{t} w_i^2 \mathrm{Var}\left( Q_{q_i}(p_i) \right) dp_i$$

$$= \frac{1}{t} \sum_{i=1}^{K} w_i^2 \int_{0}^{t} \mathrm{Var}\left( Q_{q_i}(p_i) \right) dp_i$$

$$= \frac{1}{t} \sum_{i=1}^{K} w_i^2 \int_{0}^{t} u_i b_i \, dp_i \qquad \text{Lemma 2}$$

$$= \frac{1}{t} \sum_{i=1}^{K} w_i^2 q_i \int_{0}^{s_i} u_i b_i \, dp_i \qquad s_i\text{-periodicity of } u_i \text{ and } b_i$$

$$= \frac{1}{t} \sum_{i=1}^{K} w_i^2 q_i \int_{0}^{s_i} (s_i - p_i) \, p_i \, dp_i$$

$$= \frac{1}{6t} \sum_{i=1}^{K} w_i^2 q_i s_i^3$$

$$= \frac{t^2}{6} \sum_{i=1}^{K} \frac{w_i^2}{q_i^2}$$

$$\square$$

**Lemma 4.** *Let $Q$ be a fixed-point quantizer. Assume that parameters $p_1 \ldots p_K$ are sampled from $\mathcal{U}[-t, t]$ for arbitrary $t > 0$. Then, $\min_{q_1 \ldots q_K} \mathbb{E}_{p_1 \ldots p_K}[\mathrm{Var}(e_p^{q_1 \ldots q_K})]$ subject to $Q = \sum_{i=1}^{K} q_i$ is minimized by $q_i = Q \frac{w_i^{2/3}}{\sum_{k=1}^{K} w_k^{2/3}}.$*

*Proof.* Define

$$f(\boldsymbol{q}) = \mathbb{E}_{p_1 \dots p_K}[\mathrm{Var}(e_p^{q_1 \dots q_K})]$$

$$g(\boldsymbol{q}) = \left(\sum_{i=1}^{n} q_i\right)$$

$$\mathscr{L}(\boldsymbol{q}) = f(\boldsymbol{q}) - \lambda g(\boldsymbol{q}) \text{ (Lagrangian)}$$

Any (local) minimum $\hat{\boldsymbol{q}}$ satisfies

$$\nabla \mathscr{L}(\hat{\boldsymbol{q}}) = \mathbf{0}$$

$$\iff \nabla \frac{t^2}{6} \sum_{i=1}^{K} \frac{w_i^2}{q_i^2} - \lambda \nabla \sum_{i=1}^{K} q_i = 0 \land \sum_{i=1}^{K} q_i = Q \qquad \text{Lemma 3}$$

$$\iff \forall i = 1 \dots n. \ \frac{t^2}{-3} \frac{w_i^2}{q_i^3} = \lambda \land \sum_{i=1}^{K} q_i = Q$$

$$\iff \forall i = 1 \dots n. \ q_i = \sqrt[3]{\frac{t^2}{-3\lambda} w_i^2} \land \sum_{i=1}^{K} q_i = Q$$

$$\implies \forall i = 1 \dots n. \ q_i = Q \frac{w_i^{2/3}}{\sum_{j=1}^{K} w_j^{2/3}}$$

$\square$

## B.1   PROOF OF THEOREM 1

*Proof.* Using Lemma 4, it is straightforward to show that for any $V$, $\min_{q_1 \dots q_K} \sum_{i=1}^{K} q_i$ subject to $\mathbb{E}_{p_1 \dots p_K}[\mathrm{Var}(e_p^{q_1 \dots q_K})] = V$ is minimized by $q_i = C w_i^{2/3}$ for the unique $C \in \mathbb{R}_{>0}$ that satisfies $\mathbb{E}_{p_1 \dots p_K}[\mathrm{Var}(e_p^{q_1 \dots q_K})] = V$.

Then, taking $V = \mathbb{E}_{p_1 \dots p_K}[\mathrm{Var}(e_p^q)]$ and $C = \sqrt{\frac{a}{b}}$ (see Theorem 1), we do indeed get

$$\mathbb{E}_{p_1 \dots p_K}[\mathrm{Var}(e_p^{q_1 \dots q_K})]$$

$$= \frac{t^2}{6} \sum_{i=1}^{K} \frac{w_i^2}{\left(C w_i^{2/3}\right)^2} \qquad \text{Lemma 3}$$

$$= \frac{1}{C^2} \frac{t^2}{6} \sum_{i=1}^{K} w_i^{2/3}$$

$$= \frac{\sum_{j=1}^{K} \frac{w_j^2}{q_j^2}}{\sum_{j=1}^{K} w_j^{2/3}} \frac{t^2}{6} \sum_{i=1}^{K} w_i^{2/3}$$

$$= \frac{t^2}{6} \sum_{j=1}^{K} \frac{w_j^2}{q^2}$$

$$= \mathbb{E}_{p_1 \dots p_K}[\mathrm{Var}(e_p^q)] \qquad \text{lemma 3} \qquad \square$$

