# OpenReview forum: "DAdaQuant: Doubly-adaptive quantization for communication-efficient Federated Learning"
_ICLR.cc/2022/Conference — ICLR 2022 Submitted_

### Official Review · Reviewer_2tGb · 2021-10-31

**Correctness:** 3
**Technical Novelty And Significance:** 2
**Empirical Novelty And Significance:** 2
**Recommendation:** 3
**Confidence:** 4

**Main Review:**

The strengths:

1, The paper develops a new communication-efficient quantized FL, Particularly, the client-adaptive quantization is the first time to be considered for FL algorithms.

2,  The paper has done extensive experiments to show the improvement of proposed methods.

The weaknesses:

1,  Some math notations used in this paper miss definitions. For example, the definition of  $E_{p_1, ..., p_K}[Var(Q_q(p))]$ is missing.

2, The quantization $Q_q(p)$ used in this paper based on section 3.3 QUANTIZATION WITH FEDERATED QSGD is "$Q_q(p)$ then returns the sign of p and jpj rounded to one of the endpoints". This quantization is not an unbiased estimator of $p$, which is conflict with the claim on page 14.

3, The paper introduces the expected variance of an accumulation of quantized parameter $E[Var(\sum Q(p))]$ as a measure of the performance of quantized FL algorithm and tries to minimize it in Thereom 1. The authors should explain more why it is a good measure.  Why not give the theoretical analysis in the following logic line: convergence guarantee -> get rounds T, given $epsilon$ -> get computational complexity and communication cost?

4, In algorithm 1, RunClient($c_k$) is missing between line 8 and line 9.


**Summary Of The Paper:**

The paper tries to reduce the communication cost of the Federated Learning (FL) algorithms.  They introduce the doubly-adaptive quantization algorithm (DAdaQuant), which adopts two quantization techniques: 1, time-adaptive quantization; 2, client-adaptive quantization.  The empirical studies also show that DAdaQuant can improve client sever compression.

**Summary Of The Review:**

The paper proposes an interesting communication-efficient FL algorithm. However, the theoretical analysis is not convincing and the commonly used analyses, like communication cost via communication rounds, are missing.

---

> ### Author Response · Authors · 2021-11-16
> **Response to Official Review of Paper472 by Reviewer 2tGb**
>
> We thank the reviewer for pointing out shortcomings in the formalization of our ideas and would like to rectify them below:
>
> > Some math notations used in this paper miss definitions. For example, the definition of
>  $E_{p_1\ldots p_K}[Var(Q_q(p))]$ is missing.
>
> We define $E_{p_1\ldots p_K}[Var(Q_q(p))]$ as the expectation of $Var(Q_q(p))$ over the parameters $p_1\ldots p_k$ (recall that $Q_q(p) = \sum_{i=1}^K{w_i Q_q(p_i)}$) and the internal randomness of $Q_q(p)$. Theorem 1 treats $p_1\ldots p_k$ as random variables with a uniform distribution over [0,1].
>
> We would also like to explicitly define the expected quantization error for client-dynamic quantization $e_p^{q_1\ldots q_k}$  as  $|p - Q_{q_1\ldots q_k}(p)|$, where
> $Q_{q_1\ldots q_k}(p) = \sum_{i=1}^K{w_i Q_{q_i}(p_i)}$ is the weighted average of the parameters $p_i$ that have been quantized with quantization levels $q_i$.
> .
>
> We will add these missing definitions and any others that we become aware of to our paper.
>
> > [The Federated QSGD quantization $Q_q(p)$] is not an unbiased estimator of p, which is conflict with the claim on page 14.
>
> The reviewer correctly points out that the QSGD description in Section 3.3 does not immediately imply unbiased quantization. We would like to clarify that QSGD stochastically rounds p, meaning that  p is rounded up with a probability proportional to the distance from p to the lower endpoint of its quantization interval (see Figure 7). It is straightforward to show that then, $Q_q(p)$ is in fact an unbiased estimator. We will add this statement and proof to our paper.
>
> > The paper introduces the expected variance of an accumulation of quantized parameter $E[Var(\sum Q(p))]$ as a measure of the performance of quantized FL algorithm and tries to minimize it in Thereom 1. The authors should explain more why it is a good measure.
>
> Several papers in distributed training and FL derive convergence results for quantized training
> that rely on upper bounds $b$ on the variance $Var(\sum Q(p))$, where the variance is taken over the internal randomness of the quantizer $Q$ and the convergence
> speed depends on $b$ [[Li, Hao, et al., 2017](https://arxiv.org/abs/1706.02379), [Horváth, Samuel, et al., 2019](https://arxiv.org/abs/1904.05115), [Reisizadeh, Amirhossein, et al., 2020](https://arxiv.org/abs/1909.13014)]. Our measure of the quantization is inspired by these bounds. Since our focus lies more in empirical rather than in theoretical results, we prefer the average case rather than the worst case as a measure of quantization error. To this end, we compute the average expected variance $E_{p_1\ldots p_K}[Var(Q_q(p))]$ by taking the expectation over $p_1\ldots p_k$.
>
> We note that this is merely a heuristic and by no means a rigorous theoretical analysis. While we obtain good experimental results with this metric, it is conceivable that other quantization error metrics can lower quantization costs even further or facilitate a theoretical analysis of DAdaQuant.
>
>
>
> > Why not give the theoretical analysis in the following logic line: convergence guarantee -> get rounds T, given  -> get computational complexity and communication cost?
>
> We point to a convergence result for quantized FL in Section 3.6 that can provide a basis for an in-depth theoretical analysis of DAdaQuant. However, we think that convergence guarantees offered by such proofs usually include problem-specific parameters that are hard to estimate and hence do not allow for an accurate estimate of the number of training rounds or the amount of communication required to converge. For this reason, we rely more on empirical than theoretical results to justify our algorithm.
>
> >  In algorithm 1, RunClient($c_k$) is missing between line 8 and line 9.
>
> Our intention was for  RunClient($c_k$) to be executed in parallel with RunServer($c_k$).
> We realize that in this case, RunClient is missing a while loop over its body. We thank the reviewer for spotting this inconsistency and will correct the bug in the paper.

---

### Official Review · Reviewer_d8ve · 2021-11-01

**Correctness:** 3
**Technical Novelty And Significance:** 2
**Empirical Novelty And Significance:** 3
**Recommendation:** 8
**Confidence:** 2

**Main Review:**

The paper provides clear intuition behind the double adaptive quantization approach. The work also presents solid empirical studies by comparing with a series of baseline algorithms over multiple classical datasets. The work overall is well presented.

Major concerns and questions:
1. For Theorem 1, what is the definition of $e^{p_1\cdots p_K}_q$? How is the agent-specific quantization level $q_i$'s relates to the given quantization level $q$?
2. For Figure 4, the authors comment that for DAdaQuant, the communication cost does not scale with the number of clients. The reviewer thinks that despite per iteration the communication cost does not increase, the overall number of training rounds will increase, which causes the total communication cost to grow, especially in a data heterogeneous case. Can the author comments on this?

Minor questions:
1. In section 3.5, the authors comment "We observe that Ep1...pK [Var(Qq(p))] is a useful statistic of the quantization error
because...". The reviewer does not doubt the intuition. But is there any rigorous justification for this observation?
2. Have the authors tried other orthogonal techniques combining with quantization to further improve the communication cost empirically?

**Summary Of The Paper:**

The paper proposes a communication-efficient federated learning framework named DAdaQuant. It is a quantization-based FL compression algorithm, which chooses both time-adaptive and client-adaptive quantization levels to improve the communication efficiency of FL algorithms over previous quantization works. The work provides math intuition behind choosing the client-adaptive quantization level. The authors also show solid empirical studies over FL datasets and validate the superiority of the proposed approach in communication efficiency.

**Summary Of The Review:**

The paper is lean towards empirical studies. So the reviewer does not judge from the theoretical contribution perspective. Overall, the paper well presents the intuition and the logic of the data-adaptive quantization approach. The empirical study part is well-rounded and the results are relatively solid.

---

> ### Author Response · Authors · 2021-11-16
> **Response to Official Review of Paper472 by Reviewer d8ve**
>
> We thank the reviewer for the positive comments and would like to address some of their concerns.
>
> > For Theorem 1, what is the definition of $e_q^{p_1\ldots p_k}$? How is the agent-specific quantization level $q_i$'s relates to the given quantization level $q$ ?
>
> In analogy to the definition of the static quantization error $e^q_p = |p - Q_q(p)|$, we define the adaptive quantization error  $e_p^{q_1\ldots q_k} = |p - Q_{q_1\ldots q_k}(p)| $, where
> $Q_{q_1\ldots q_k}(p) = \sum_{i=1}^K{w_i Q_{q_i}(p_i)}$ is the weighted average of the parameters $p_i$ that have been quantized with quantization levels $q_i$.
>
> We use the given quantization level $q$ to compute the expected quantization error $e^q_p$ when doing static quantization with quantization level $q$. We then use $e^q_p$ as a “target” quantization error that we "are allowed to" incur. Theorem 1 then tells us among all choices of $q_i$ for client-adaptive quantization, the optimal choice (w.r.t. lowest communication costs) that has an expected quantization error $e_p^{q_1\ldots q_k}$ not exceeding $e_p^q$.
>
> We will clarify this part further in an improved version of our paper.
>
> > The reviewer thinks that despite per iteration the communication cost does not increase, the overall number of training rounds will increase, which causes the total communication cost to grow, especially in a data heterogeneous case.
>
> Figure 4 demonstrates how our approach works differently to a comparable quantization scheme designed for a full participation FL setup. We agree with the reviewer that a partial participation federated learning scenario (only a subset of the devices upload gradients to the server) may increase the number of training rounds. However, in practice, we observe that a full participation method like AdaQuantFL requires similarly many training rounds to converge as
> our partial participation algorithm (see https://postimg.cc/FdCp3QPv).
>
> It is also true that the training rounds may increase when we have a greater number of clients, however, we believe our proposed method has the ability to automatically (by adaptively adjust quantization levels) match the performance of uncompressed models, as illustrated in Table 1 of our paper. In this case, our proposed method also provides communication savings compared to canonical FL training.
>
> > In section 3.5, the authors comment "We observe that Ep1...pK [Var(Qq(p))] is a useful statistic of the quantization error because...". The reviewer does not doubt the intuition. But is there any rigorous justification for this observation?
>
> Several papers in distributed training and FL derive convergence results for quantized training
> that rely on upper bounds $b$ on the variance $Var(\sum Q(p))$ [[Li, Hao, et al., 2017](https://arxiv.org/abs/1706.02379), [Horváth, Samuel, et al., 2019](https://arxiv.org/abs/1904.05115), [Reisizadeh, Amirhossein, et al., 2020](https://arxiv.org/abs/1909.13014)], where the variance is taken over the internal randomness of the quantizer $Q$ and the convergence speed depends on $b$. Our measure of the quantization is inspired by these bounds. Since our focus lies more in empirical rather than in theoretical results, we prefer the average case rather than the worst case as a measure of quantization error. To this end, we compute the average expected variance $E_{p_1\ldots p_K}[Var(Q_q(p))]$ by taking the expectation over $p_1\ldots p_k$.
>
> We note that this is merely a heuristic and by no means a rigorous theoretical analysis. While we obtain good experimental results with this metric, it is conceivable that other quantization error metrics can lower quantization costs even further or facilitate a theoretical analysis of DAdaQuant.
>
> > Have the authors tried other orthogonal techniques combining with quantization to further improve the communication cost empirically?
>
> No. However, we believe that quantization error correction via error accumulation is a low-hanging fruit that may allow for even more aggressive quantization. In addition, it is conceivable to vary quantization levels along additional dimensions, for example different layers of a neural network.

---

> > ### Comment · Reviewer_d8ve · 2021-12-06
> > **Thank you for your response**
> >
> > Thank you for your response!

---

### Official Review · Reviewer_eicQ · 2021-11-02

**Correctness:** 3
**Technical Novelty And Significance:** 3
**Empirical Novelty And Significance:** 3
**Recommendation:** 5
**Confidence:** 3

**Main Review:**

Strength:

(1) A novel double quantization design

(2) Communication overhead saving is promising


Weakness

**(1)  Computational overhead in quantization**

This paper proposes a double quantization strategy for efficient FL.  While the saving in the communication overhead is promising, there is little discussion on the extra computational overhead introduced by the algorithm. For instance: i) what is the complexity of performing the time-adaptive and  client-adaptive quantization algorithm, ii) how does the overall training time being affected if we use the proposed algorithm, iii) what is the ratio between the communication time saving and the total training time

**(2)  Comparison with other communication efficient algorithms**

This paper compares with quantization baselines. How does the double quantization algorithm perform when we compare it with sketching-based FL methods?



**Summary Of The Paper:**

This paper studies the federated learning problem with a focus on communication efficiency. The major contribution of this paper can be summarized as:

(1) Propose a time-adaptive quantization algorithm that adjusts the quantization level
as training progresses

(2) Propose a client-adaptive quantization algorithm that assigns quantization level to individual clients

**Summary Of The Review:**

This paper is well-organized with a clear presentation. However,  there exist concerns regarding the efficiency of the proposed algorithm

The authors are strongly encouraged to address these shortcomings by:

(1) Benchmarking the total training time and the computational overhead in quantization

(2) Comparison with other  communication efficient  FL strategies such as sketching

---

> ### Author Response · Authors · 2021-11-16
> **Response to Official Review of Paper472 by Reviewer eicQ**
>
> We thank the reviewer for their time and the important questions raised regarding DAdaQuant’s computational efficiency and performance compared to Sketching-based compression.
>
> **Computational overhead in quantization**
>
> > What is the complexity of performing the time-adaptive and client-adaptive quantization algorithm? How does the overall training time being affected if we use the proposed algorithm?
>
> For our FEMNIST experiments, one training round took 36 seconds on a client.
>
> During this round, time-adaptive quantization executes a negligible $O(c)$ arithmetic operations on the server, where $c$ is the number of sampled clients (10 for our experiments). This took < 1ms. On each client, time-adaptive quantization performs 1 additional evaluation epochs per training round, which is a small fraction of the 20 training epochs per training round. This took 0.17 seconds.
>
> Client-adaptive quantization only executes a negligible $O(c)$ arithmetic operations on the server. This took < 1ms.
>
> To compute the quantized weights, our (unoptimized, cpu-based) quantization implementation took 0.24 seconds. The original QSGD paper [(Alistarh et al., 2017)](https://arxiv.org/abs/1610.02132) also confirms that the computational overhead of gradient compression is minor, which cross-validates our implementation.
>
> Hence, for this experiment, we observe a computational overhead of ~1.2% compared to uncompressed training.
>
> > What is the ratio between the communication time saving and the total training time?
>
> Unfortunately, it is notoriously difficult to quantify savings in communication time with respect to the total training time because bandwidth, as well as compute power, can differ by several orders of magnitude for different FL setups. This is why we only report savings in the amount of communication. We note that in addition to reducing the overall training time, less communication also means significantly lower energy consumption [(Qiu et al., 2020)](https://arxiv.org/abs/2102.07627) and less monetary costs (e.g. for mobile phone connections).
>
> Comparison with other communication efficient algorithms
>
> > How does the double quantization algorithm perform when we compare it with sketching-based FL methods?
>
> Our empirical evaluations focus on quantization-based algorithms because they are directly comparable to our method, and because, to the best of our knowledge, they achieve the best compression for FL.
>
> To address your question, we have now conducted another experiment with Counting-Sketch based compression that follows [FetchSGD](https://arxiv.org/abs/2007.07682).
> In our preliminary results, Sketching does not exceed 60% accuracy for FEMNIST with a compression factor of just 2x. In comparison, DAdaQuant achieves over 77% accuracy with a compression factor of over 4000x. While FetchSGD reports a 50x compression factor with nearly 80% accuracy, we hypothesize that this severe discrepancy is due to different FL setups: FetchSGD communicates gradients after every local training step, while we train locally for 20 epochs before sending parameter updates to the server. This suggests that Sketching may not be viable in a FL setup with prolonged local training. However, due to our limited experience
>  with Counting Sketches, we acknowledge that our Sketching implementation may not be optimal / entirely correct and have contacted the authors of FetchSGD for clarification.

---

### Official Review · Reviewer_CKYb · 2021-11-02

**Correctness:** 3
**Technical Novelty And Significance:** 3
**Empirical Novelty And Significance:** 3
**Recommendation:** 5
**Confidence:** 4

**Main Review:**

### Strengths
1. The proposed two adaptive quantization strategies are simple and easy to implement in practice. In the experiments, they exhibit better performance than static quantization schemes.
2. The client-level adaptive quantization scheme is new and has not appeared in literature. The idea itself makes sense and the authors proposed a theory-grounded approach to make it work.

### Weaknesses
I think the "time-adaptivity" part in the paper is a bit improper. And the authors tend to oversell their contribution. The contribution in this part is kind of trivial compared to previous works.

- In the introduction, the authors mentioned that "we observe that early training rounds can use a lower q without affecting convergence". But in fact, this observation was first made by previous literature, such as (Jhunjhunwala et al. ICASSP 2021). More generally, the intuition that one need more communication towards the end of training already appeared in two years ago in (Wang & Joshi, MLSys 2019, "Adaptive communication strategies to achieve the best error-runtime trade-off in local-update SGD"). The way the authors wrote this part can make people think this paper find this idea, which is not. The authors are supposed to make it clear in introduction that "previous literature observed that.. and we further improve their algorithms by overcoming/addressing.."
- The comparison of time-adaptive quantization schemes with previous work (Jhunjhunwala et al. ICASSP 2021) is unfair. Basically, (Jhunjhunwala et al. ICASSP 2021) let all clients to participate into each round of training because they consider the full participation FL. But the algorithm itself can be easily extended to the case where we only sample few clients at each round. One can simply replace the global loss by the average loss within the current subset. Compared to this work, the authors should demonstrate (1) moving average of the loss help; (2) a step-increase strategy is better than the strategy used in (Jhunjhunwala et al. ICASSP 2021); (3) the proposed strategy converges faster than any other static quantization schemes, as predicted in Figure 2.
- Related to the above point, I feel the authors failed to demonstrate the effectiveness of the time-adaptive quantization methods. There are still many remaining questions. In their experiments, the authors do not show whether the moving average of loss help. The authors do not compare the results with previous works AdaQuantFL. The authors do not show the proposed scheme is faster than any other static quantization schemes, as predicted in Figure 2. Instead, they just fix one quantization level and report the final accuracy.
- In addition, the authors claim Pareto optimality in their experiments. I don't agree with this. Although the proposed algorithm is better, why is it the optimal one?


**Summary Of The Paper:**

This paper studies how to compress the local model changes in federated learning to save uplink communication costs. In particular, the authors found (1) adaptively increasing the number of quantization levels (2) adaptively assigning different quantization levels on different clients, can effectively outperform previous static quantization schemes. They evaluated the proposed algorithm on multiple federated learning datasets.

**Summary Of The Review:**

In general, I think the client-adaptive quantization part is quite interesting and novel. But the time-adaptive quantization part is incomplete. The authors fail to demonstrate the effectiveness of their proposed method.

---

> ### Author Response · Authors · 2021-11-16
> **Response to Official Review of Paper472 by Reviewer CKYb**
>
> We thank the reviewer for the thorough feedback.
>
> > The authors are supposed to make it clear in introduction that "previous literature observed that.. and we further improve their algorithms by overcoming/addressing.."
>
>   In Section 2, we state that “AdaQuantFL introduces time-adaptive quantization to FL…”. However, we realize that this statement does not explicitly acknowledge prior findings on the required quantization levels at different times during training. We will clarify this in our paper and add a paragraph that properly attributes the observation that “early training rounds can use a lower q without affecting convergence”.
>
>
> > The comparison of time-adaptive quantization schemes with previous work (Jhunjhunwala et al. ICASSP 2021) is unfair. Basically, (Jhunjhunwala et al. ICASSP 2021) let all clients to participate into each round of training because they consider the full participation FL. But the algorithm itself can be easily extended to the case where we only sample few clients at each round. One can simply replace the global loss by the average loss within the current subset.
> Compared to [(Jhunjhunwala et al. ICASSP 2021)], the authors should demonstrate (1) moving average of the loss help; (2) a step-increase strategy is better than the strategy used in (Jhunjhunwala et al. ICASSP 2021); (3) the proposed strategy converges faster than any other static quantization schemes, as predicted in Figure 2.
>
>   In our initial experiments with time-adaptive quantization strategies, we found the average loss  within the current subset (let’s call it the “sample loss”) to be too noisy on some datasets to efficiently set the quantization level. Thus, once we became aware of AdaQuantFL (Jhunjhunwala et al. ICASSP 2021), we deemed replacing the global loss with the sample loss in AdaQuantFL as a significant change that may be considered a new algorithm rather than an extension. Rather than investigating this algorithm, we decided to introduce a new time-adaptive quantization algorithm that showed promising results in our experiments (DAdaQuant_time).  We do not claim that DAdaQuant_time is the best choice, but merely that it improves over non-time-adaptive quantization. With that said, we have extended some of our experiments to respond to (1), (2) and (3):
> * (1): A single run of DAdaQuant_time without a moving loss average on Shakespeare, a dataset with noisy loss, achieved an accuracy of 49.96% using 26.5 MB. In comparison, the same run with a moving loss average achieved an accuracy of 49.9% using 22.9 MB. This corresponds to a communication reduction by 13.4%.
>
> * (2): A single run of AdaQuantFL on FEMNIST achieved an accuracy of 78.7%, using 50.5 MB. In comparison, the same run with DAdaQuant_time achieved an accuracy of 78.4%, using 27.5 MB. This does not clearly show that DAdaQuant_time is better than AdaQuantFL, but suggests that DAdaQuant_time is a viable alternative.
>
> * (3): We do not intend to give any speed-of-convergence guarantees for DAdaQuant_time; Figure 2 is merely supposed to convey some intuition about its potential benefits.  In our empirical results, we observe that DAdaQuant_time nearly always lowers communication cost while matching the accuracy of static quantization.
>
> > The authors do not show the proposed scheme is faster than any other static quantization schemes, as predicted in Figure 2. Instead, they just fix one quantization level and report the final accuracy.
>
> For each dataset, we performed a grid search over different static quantization levels and reported static quantization results for the best one. This means that the superior performance of DAdaQuant_time over static quantization in our experiments does not depend on the static quantization level. To show that our results hold for arbitrary accuracies, we re-plot Figure 5 for DAdaQuant_time (see https://postimg.cc/8fLrXPnS) This shows that DAdaQuant_time outperforms static quantization for nearly any amount of communication / any target accuracy.
>
> > In addition, the authors claim Pareto optimality in their experiments. I don't agree with this. Although the proposed algorithm is better, why is it the optimal one?
>
> Our paper claims “[...] DAdaQuant is Pareto optimal for the datasets we have explored.” By this, we mean that for any fixed amount of communication, and for any of our datasets, DAdaQuant achieves a higher accuracy than the best static quantization algorithm that we are aware of, namely Federated QSGD. We realize that this may not be the universal understanding of Pareto optimality and will replace the Pareto optimality claim with our definition above.

---

> > ### Comment · Reviewer_CKYb · 2021-11-29
> > **Thanks for the response**
> >
> > Thank the authors for the response! I want to keep my original score as my main concern has not been addressed.
> >
> > In particular, I don't think replacing the full loss with a sampled loss in AdaQuantFL makes a new algorithm. This is just a simple extension in the case of client sampling and is not sufficient to publish a paper. Basically, the original AdaQuantFL is proposed for full client participation. So in order to apply it to the client sampling case, one must make some modifications. Using a sampled loss is a very natural choice. Just like in FedAvg, when there is client sampling, we just need to average the local model changes of the sampled clients instead of all. The algorithm is still FedAvg, as its core doesn't change.
> >
> > Although this paper contains other contributions, I feel the time-adaptive part is not enough. In the response, the authors provide some new experimental results but I didn't find them in the main paper. More importantly, as an empirical paper, there should be more discussions or ablation studies in the paper to help readers to get insight into why moving averaged loss and step-increase adaptive strategy improve previous methods instead of just posting the numbers. To incorporate this suggestion, I'm afraid the paper needs a significant revision.
> >
> > I also have a minor question, in the experiment (2), to demonstrate the effectiveness of step-increase decay, do you fix the loss computation mechanism in  AdaQauntFL and DAdaQuant the same? I'm not sure whether the improvement come from the moving averaged loss or step-increase strategy.

---

> > > ### Author Response · Authors · 2021-11-30
> > > **Response to Reviewer CKYb**
> > >
> > > We thank Reviewer CKYb for their response.
> > >
> > > > Although this paper contains other contributions, I feel the time-adaptive part is not enough.
> > >
> > > We would like to ask the reviewer to take another look at our `Contributions` paragraph. We deliberately do not claim any novelty for time-adaptive quantization, but only for quantization along a new adaptive dimension (clients) and for the combination of time-adaptive quantization with client-adaptive quantization.
> > >
> > > > In the response, the authors provide some new experimental results but I didn't find them in the main paper.
> > >
> > > We have added the experiments in question to appendices A.3 and A.5.
> > >
> > > > I also have a minor question, in the experiment (2), to demonstrate the effectiveness of step-increase decay, do you fix the loss computation mechanism in AdaQauntFL and DAdaQuant the same?
> > >
> > > Yes. All our experiments compute the loss in the same way.

---

### Decision · Program_Chairs · 2022-01-20

**Decision:**

Reject

**Comment:**

Dear authors,

I apologize to the authors for insufficient discussion in the discussion period. Thanks for carefully responding to reviewers. Nevertheless, I have read the paper as well, and the situation is clear to me (even without further discussion). I will not summarize what the paper is about, but will instead mention some of the key issues.

1) The proposed idea is simple, and in fact, it has been known to me for a number of years. I did not think it was worth publishing. This on its own is not a reason for rejection, but I wanted to mention this anyway to convey the idea that I consider this work very incremental.
2) The idea is not supported by any convergence theory. Hence, it remains a heuristic, which the authors admit. In such a case, the paper should be judged by its practical performance, novelty and efficacy of ideas, and the strength of the empirical results, rather than on the theory. However, these parts of the paper remain lacking compared to the standard one would expect from an ICLR paper.
3) Several elements of the ideas behind this work existed in the literature already (e.g., adaptive quantization, time-varying quantization, ...). Reviewers have noticed this.
4) The authors compare to fixed / non-adaptive quantization strategies which have already been surpassed in subsequent work. Indeed, QSGD was developed 4 years ago. The quantizers of Horvath et al in the natural compression/natural dithering family have exponentially better variance for any given number of levels. This baseline, which does not use any adaptivity, should be better, I believe, to what the author propose. If not, a comparison is needed.
5) FedAvg is not the theoretical nor practical SOTA method for the problem the authors are solving. Faster and more communication efficient methods exist. For example, method based on error feedback (e.g., the works of Stich, Koloskova and others), MARINA method (Gorbunov et al), SCAFFOLD (Karimireddy et al) and so on. All can be combined with quantization.
6) The reviewer who assigned this paper score 8 was least confident. I did not find any comments in the review of this reviewer that would sufficiently justify the high score. The review was brief and not very informative to me as the AC. All other reviewers were inclined to reject the paper.
7) There are issues in the mathematics - although the mathematics is simple and not the key of the paper. This needs to be thoroughly revised. Some answers were given in author response.
8) Why should expected variance be a good measure? Did you try to break this measure? That is, did you try to construct problems for which this measure would work worse than the worst case variance?

Because of the above, and additional reasons mentioned in the reviewers, I have no other option but to reject the paper.

Area Chair